# Understanding the Role of Innate Immune Cells and Identifying Genes in Breast Cancer Microenvironment

**DOI:** 10.3390/cancers12082226

**Published:** 2020-08-09

**Authors:** Israa Shihab, Bariaa A. Khalil, Noha Mousaad Elemam, Ibrahim Y. Hachim, Mahmood Yaseen Hachim, Rifat A. Hamoudi, Azzam A. Maghazachi

**Affiliations:** 1Department of Clinical Sciences and the Immuno-Oncology Group, Sharjah Institute for Medical Research, University of Sharjah, Sharjah 27272, UAE; U18105557@sharjah.ac.ae (I.S.); U19106038@sharjah.ac.ae (B.A.K.); noha.elemam211@gmail.com (N.M.E.); ihachim@sharjah.ac.ae (I.Y.H.); rhamoudi@sharjah.ac.ae (R.A.H.); 2College of Medicine, Mohammed Bin Rashid University of Medicine and Health Sciences, Dubai 505055, UAE; U16101425@sharjah.ac.ae

**Keywords:** innate immunity, breast cancer, microenvironment, NK cells, bioinformatics, genes, immune cells, cytokines, chemokines

## Abstract

The innate immune system is the first line of defense against invading pathogens and has a major role in clearing transformed cells, besides its essential role in activating the adaptive immune system. Macrophages, dendritic cells, NK cells, and granulocytes are part of the innate immune system that accumulate in the tumor microenvironment such as breast cancer. These cells induce inflammation in situ by secreting cytokines and chemokines that promote tumor growth and progression, in addition to orchestrating the activities of other immune cells. In breast cancer microenvironment, innate immune cells are skewed towards immunosuppression that may lead to tumor evasion. However, the mechanisms by which immune cells could interact with breast cancer cells are complex and not fully understood. Therefore, the importance of the mammary tumor microenvironment in the development, growth, and progression of cancer is widely recognized. With the advances of using bioinformatics and analyzing data from gene banks, several genes involved in NK cells of breast cancer individuals have been identified. In this review, we discuss the activities of certain genes involved in the cross-talk among NK cells and breast cancer. Consequently, altering tumor immune microenvironment can make breast tumors more responsive to immunotherapy.

## 1. Introduction

### 1.1. Breast Cancer Heterogeneity and the Role of Immune Cells

Among all types of cancer, breast cancer is the most frequent type in women worldwide. Each year, this malignancy affects more than two million women globally, which is the primary cause of death [1]. In 2018, it is estimated that 627,000 women died from breast cancer, approximately 12% of all cancer deaths among women [2]. Despite the advances in management strategies in diagnosis and treatment, the high heterogeneity of this cancer makes it difficult to treat. Breast cancer (BC) can be histologically classified to luminal A, luminal B, Her2 positive, basal-like, and normal-like, with respect to the biomarkers status of estrogen receptor (ER), progesterone receptor (PR), and HER2 [3,4]. It has been found that triple negative breast cancer is the most aggressive and has the lowest overall survival rate between other types of breast cancer [5]. Besides, the tumor microenvironment (TME) plays a dual role in tumor progression and immune repression. The tumor microenvironment influences immune cells to be activated and differentiated towards enhancing tumor progression and favors inhibition of anti-tumor activities. However, the complete interaction between breast cancer cells and the immune cells in the tumor microenvironment is not very well understood [6,7].

The cells of the immune system that are implied in the immune defense are components of either the innate or the adaptive immune system. The first early response to a pathogen or transformed cells is a function of the innate immune cells that act rapidly and non-specifically to an attack to prevent spreading of the foreign pathogen [8]. The cells of the innate immune system control and clear invasion via various mechanisms, such as release of cytotoxic molecules, engagement of more immune cells, complement pathway activation, or activation of the phagocytosis process [9]. In cancer, tumor cells may release some cellular components that may activate the innate immune cells, which in turn establish antitumor immunity in the microenvironment and hence induce tumor eradication [10,11].

The activation of innate immune cells results in the interplay of events that work together to control and destroy tumor cells. For instance, upon encounter with transformed tumor cells, some immune cells such as macrophages and dendritic cells become activated and release large amounts of pro-inflammatory cytokines such as IL-12, IL-15, and type 1 interferon (IFN) that activate natural killer (NK) cells and T helper cell differentiation [12,13]. The activation of innate cells in turn stimulates adaptive immune system by releasing large amounts of IFN-γ and chemokines including CCL3 and CCL4 [14,15,16]. In addition, the release of IFN-γ and IL-12 is necessary to destroy tumor cells by switching anti-inflammatory M2 to M1 phenotype, along with an increase of MHC I molecules on tumor cells [17,18]. Therefore, understanding the cross talk among these cell types and tumor cells can advance the knowledge of how immune cells infiltrate into the tumor microenvironment, which may help in choosing the best personalized immunotherapy therapeutic approaches.

### 1.2. Components of the Tumor Microenvironment

Apart from tumor cells present in breast cancer, the tumor microenvironment (TME) contains other cells such as fibroblasts and immune cells, extracellular matrix (ECM), blood vessels, and some signaling molecules [19]. TME is important for tumor angiogenesis, immune cells’ inhibition, escape from immune surveillance, as well as tumor proliferation and survival [7]. ECM plays a major role in breast cancer microenvironment, as it provides the physical support to solid tumors by its protein components such as collagen and proteoglycan. Additionally, ECM is rich in several growth factors that can regulate angiogenesis and inflammation [20]. In advanced stages, ECM becomes disrupted and dysregulated allowing tumor growth by metalloproteinases (MMPs), aiding in cells migration from or into the TME, and thus assisting tumor metastasis [21,22]. 

In addition to ECM disruption, rapid growing tumors lack sufficient blood supply, creating a hypoxic and acidic environment. This environment will induce tumor cells to release vascular endothelial growth factor (VEGF) and other angiogenic molecules, causing the formation of new blood vessels from the existing vessels for the tumor [23,24], as shown in Figure 1. Furthermore, the newly formed vessels have irregular wall layers, making them leaky with disrupted blood flow, thus providing inadequate blood and nutrient supply that further enhances angiogenesis and tumor growth [25,26]. 

TME makes a niche for the proliferation of tumor cells and the neighboring different types of cells such as endothelial and immune cells. The interaction between cancer cells and immune cells promotes modifications in the tumor microenvironment, which induce tumor angiogenesis, growth, progression, and metastasis [27,28]. Moreover, immune cells secrete chemokines and cytokines that attract other cells into the microenvironment, consequently contributing to tumor progression [29]. The immune cells could inhibit or promote tumor growth and their possible dual roles have been previously investigated intensively. For instance, immune suppressive TME has been reported to be essential for tumor survival and growth. Indeed, the importance of immune cells is illustrated by the fact that chronic inflammation correlates with a higher chance of tumor incidence [30]. For this reason, innate immune cells present in the tumor microenvironment such as macrophages, myeloid-derived suppressive cells (MDSC), and neutrophils are associated with immunosuppression, poor prognosis, and tumor progression [31]. On the other hand, the presence of other innate immune cells such as NK cells in TME indicates good prognosis and tumor clearance [6]. Therefore, understanding the TME development and its components, as well as the interaction between various cell types in TME is quite crucial for development and progress of breast cancer (Figure 1).

### 1.3. Effects of Innate Immune Cells on Breast Cancer Microenvironment

The effect of the immune cells on breast cancer growth and survival depends on both the subtype of cancer and the presence of inflammatory cells in the breast cancer microenvironment [32]. Breast cancer cells interact with various types of immune cells, among which are innate immune cells including monocytes, macrophages, NK cells, and neutrophils. These immune cells can regulate tumor cells proliferation, development, and progression. In the initial stages, continuous cell death occurs at a great pace causing an inhibition of the immune system present in the tumor microenvironment. Additionally, the clearance of the dead cells by the phagocytosis pathway may create an anti-inflammatory environment which further inhibits the local immune system [17,33]. On the contrary, inflammation could be induced by innate immune cells upon encounter with transformed cells. For example, upon contact with tumor cells, macrophages become activated, thus releasing inflammatory cytokines and chemokines such as TNF, IL-1, IL-6, IL-8, and IL-12 [17,34,35]. In addition, macrophages produce reactive oxygen species, nitrogen oxide species, and different growth factors promoting the generation of an inflammatory environment. Other studies have reported that macrophages could suppress the activity of anti-tumor immune cells and promote tumor progression, survival, and growth [36], as well as stimulate MDSCs to secrete the anti-inflammatory cytokine IL-10 [37]. Many clinical studies have shown that macrophages may increase the metastasis of breast cancer into the lungs, which correlates with poor prognosis of this tumor [38]. Moreover, recruitment of tumor associated macrophages (TAMs) in the cancer environment may promote immunosuppression, angiogenesis, and cancer protection from the cytokine-induced cell death due to activation of Akt pathway [39]. It is worth mentioning that lymphocytes play a crucial function in early tumor immune surveillance, via recognizing and killing early tumor cells, hence preventing breast cancer tumorigenesis [40].

NK cells act as anti-tumor effectors by producing cytotoxic molecules which lyse tumor cells or by secreting large amounts of cytokines and chemokines that can activate and recruit cells of the adaptive immune system [41,42,43,44]. NK cells become activated upon cross-linking and binding of their activating receptors and the target tumor cells, thus increasing the threshold of activation signals above the inhibitory signals [45]. NKG2D is an essential NK activating receptor that can recognize tumor cells via binding to stress ligands (MICA, MICB, and ULBPs) expressed by transformed cells. However, tumor cells try to evade NK cell activity by shedding these NKG2D ligands in order to obstruct NK cells from recognizing tumor cells and performing their cytolytic activity, thus resulting in the escape of breast cancer cells [46,47]. 

## 2. Role of Innate Cells in Breast Cancer

### 2.1. Role of Dendritic Cells in Breast Cancer

Enhancing the anti-tumor functional activity of CD8^+^ T cells is central to most immunotherapies. Dendritic cells (DCs) are crucial innate antigen presenting cells (APCs) that aid in the licensing of antigen-specific naïve T-cells for priming, maturation, and differentiation. This occurs by the means of two signals: (1) presenting antigens to T cells; and (2) engagement of co-stimulatory receptors such as CD80 and CD86. DCs have a chief anti-tumor function as they cause T-cell activation and aid in their abilities to fight cancer. As a result, any alteration or impairment of DCs by the tumor microenvironment would be risky by allowing cancer to escape the immune system and progress towards metastasis [48]. 

Conventional dendritic cells (cDCs) are considered the central inducers of T-cell activation and maturation due to their antigen presenting capacity. In mice, tumor-migrating DCs are thought to be required for de novo T cell priming in the lymph nodes, which can be divided based on specific markers into two lineages: the minor CD8^α^/CD103^+^ cDC1 population that depends on the transcription factors interferon regulatory factor (IRF) 8 and basic leucine zipper transcription factor ATF-like 3 (BATF3) and the predominant CD11b^+^ cDC2 population relying on the transcription factor IRF4 [49,50].

The recruitment of CD103^+^ cDCs into tumors is necessary for the development of a CD8^+^ T-cell response in order to mediate the efficacy of adoptive cell transfer therapy [50,51]. Moreover, intra-tumoral CD103^+^ subset of DCs in murine models of triple-negative and luminal B models were evaluated for the expression of immune regulators TIM-3, as it is a potential therapeutic target. TIM-3 expression was predominantly localized to myeloid cells in both human and murine tumors [52]. Anti-TIM-3 antibody showed an improvement of response to paclitaxel chemotherapy with no evidence of toxicity, with an upregulation of CXCL9 within intra-tumoral DCs [52]. However, this therapeutic efficacy was ablated by blockage of its respective chemokine receptor CXCR3 or deficiencies of BATF3 or IRF8 [52]. Furthermore, this combinational therapy enhanced CD8^+^ T-cell-dependent activity as well as increased Granzyme B expression [52]. 

Plasmacytoid dendritic cells (pDCs) represent a unique subset of DCs that trigger the production of large amounts of type I interferons after viral infections. Additionally, toll-like receptors (TLR7 and TLR9) on pDCs were shown to directly kill breast cancer cells through TNF-related apoptosis-inducing ligand (TRAIL) and Granzyme B in a breast cancer mouse model and ultimately inhibit breast tumor growth by initiating the sequential activation of NK cells and CD8^+^ T cells [53]. However, it has been found that pDCs infiltrating breast tumors possess an impairment of interferon-α (IFN-α) production, resulting in local regulatory T-cells amplification [54]. Besides, the production of IFN-β and TNF-α by TLR-activated healthy pDC is impaired by the breast tumor environment. This might be attributed to TGF-β and TNF-α as major soluble factors involved in the functional alteration of tumor associated pDCs, via inhibition of IRF-7 expression and nuclear translocation in pDC after their exposure to tumor-derived supernatants or recombinant TGF-β1 and TNF-α [55]. Hence, restoring the ability of tumor associated pDCs to produce IFN-α in combination with TLR7/9-based immunotherapy with TGF-β and TNF-α antagonists can restore the antitumor immunity in breast cancer patients [55]. Induction of TLRs in macrophages leads to their early and late spreading via MAP kinase p38 and MyD88 [56]. Therefore, it might be that some of the ligands related to cellular pathways may lead to the activation of more than one type of innate immune cells, whereas others may activate one specific type. This can make the immune response selective or specific depending on the initial trigger providing it with greater plasticity. However, this plasticity may sometimes be exploited by the tumor cells. 

Although DC subsets are implicated in killing BC cells, a study on 152 patients with invasive non-metastatic BC showed that tumor infiltrated by mature CD208/DC-LAMP^+^ DCs and CD3^+^ T cells were strongly correlated with lymph node involvement and tumor grade [57]. The presence of CD123^+^ pDC in primary tumors was strongly associated with shorter overall survival and relapse-free survival, as well as being an independent prognostic factor for overall and relapse-free survival [57]. The mRNA levels of 730 immune-related genes using nanostring technology in primary and metastatic cancer samples were also evaluated. Metastatic tumors showed a coordinated downregulation of chemoattractant ligand/receptor pairs (CCL19/CCR7, CXCL9/CXCR3, and IL15/IL15R), interferon regulated genes (STAT1; IRF-1, IRF-4, and IRF-7; and IFI-27 and IFI-35), granzyme/granulysin, MHC class I, and immune proteasome (PSMB-8, PSMB-9, and PSMB-10) expression [58]. Furthermore, the expression of macrophage markers (CD163, CCL2/CCR2, CSF1/CSFR1, and CXCR4/CXCL12), pro-tumorigenic TLR pathway genes (CD14; TLR-1, TLR-2, TLR-4, TLR-5, and TLR-6; and MyD88), HLA-E, ecto-nuclease CD73/NT5E, and inhibitory complement receptors (CD59, CD55, and CD46) were elevated in metastasis [58]. Consequently, this study highlighted the importance of immune cells biomarkers in determining tumor characteristics and staging as well as predicting survival and relapse rates. Based on that, understanding the role of various markers and potential targets of different DC subsets in different BC types may help to develop new strategies for manipulating their function to induce anti-tumor immunity.

#### Effects of Tumor Phenotypic Markers on Dendritic Cells Anti-Tumor Activity

Studies have shown that markers on tumor cells impact the anti-tumor activity of DCs. Among these markers, the CD47 molecule signals malignant cells to escape the immune system. As with most solid tumors and hematological malignancies, blocking the CD47/SIRP-α axis between tumor cells and innate immune cells increases tumor recognition and cell phagocytosis in breast cancer [59]. In addition, cytotoxic T lymphocyte-associated antigen 4 (CTLA-4) on tumor cells impacts the function of DCs. CTLA-4 downregulates T-cell activation and inhibits its anti-tumor immune response by binding to co-stimulatory molecules on DCs (CD40, CD80, and CD86) and suppressing their ability in presenting foreign or tumor antigens [60]. A study showed a decrease in the expression of HLA-DR, CD83 and other costimulatory molecules such as CD40, CD80 and CD86 on LPS-stimulated human DCs when co-cultured with CTLA-4^+^ BC cells. In turn, suppressed DCs inhibited the proliferation of allogeneic CD4^+^/CD8^+^ T-cells, differentiation of Th1, and, consequently, the function of cytotoxic T lymphocytes “CTLs” [61]. Thus, CTLA-4 blockade therapy reverses the function of DCs and restores their anti-tumor activity.

Another mechanism by which tumors could evade the immune system is through manipulating the maturation and apoptosis of monocyte-derived DCs [62]. Tumors generate immunosuppressive factors that halt the maturation of CD34^+^ stem cells into DCs, thus leading to defects in the maturation and/or function of human monocyte-derived DCs [62]. In response to exposure of tumor culture supernatant (TSN), CD14^+^ cells increased the expression of APC surface markers, upregulated nuclear translocation of RelB, and developed allostimulatory activity. However, TSN-exposed DCs lacked the capacity to produce IL-12, rapidly underwent apoptosis and did not acquire full allostimulatory activity [62]. These effects of TSN appeared to be specific on DCs and irreversible by antibodies against known DC regulatory factors including IL-10, VEGF, TGF-β, or PGE2 [62].

### 2.2. Role of Monocytes/Macrophages in Breast Cancer

Tumor-associated macrophages (TAMs) are among the most abundant immune cells in the tumor microenvironment, which have been shown to be linked to poor prognosis and therapeutic resistance [63]. It was demonstrated ex vivo that BC cells secret factors that modulate macrophage differentiation toward the M2 phenotype and correlate with recurrence-free survival, therapeutic efficacy, and patient outcome [64]. Clinically, high numbers of CD163^+^ M2 macrophages were strongly associated with poor differentiation, rapid proliferation, ER negativity, and histological ductal type in human primary breast tumors [64].

Various complex signals and cytokines are produced by the tumor microenvironment that shape the phenotype and function of TAMs. For instance, IL-4 and IL-6 are capable of altering the human macrophage transcriptome, causing human monocyte-derived macrophages (hMDMs) to be more tumorigenic [65]. Chemokine ligand CCL18, TGFα, and CD274 (programmed cell death ligand 1, PD-L1) are among the synergistically induced genes which were found to bind to transcription factors of the signal transducer and activator of transcription (STAT) family, STAT3 and STAT6, leading to the induction of the basic leucine zipper ATF-like transcription factor (BATF) [65]. Functionally, IL-4 and IL-6 increased tumor cell motility of BC cell lines, MCF-7 and MDA-MB 231, upon co-culture with hMDMs [65].

In a model of orthotopically introduced 4T1 BC cells, another cytokine, IL-1β, was studied, where its deficiency led to tumor regression in comparison to wild-type (WT) mice with tumor growth and metastasis along with heavy infiltration of the tumors by macrophages [66]. This observation was possibly explained by the low levels of the chemokine CCL2 and colony-stimulating factor-1 (CSF-1) in IL-1β-deficient mice, hindering the recruitment of monocytes and inhibiting their differentiation into macrophages. Additionally, macrophages are considered the main source of IL-10 in WT mice which cause immunosuppression and lead to tumor progression [66]. Moreover, such IL-1β-deficient mice exhibited higher numbers of CD11b^+^ DCs secreting IL-12 in the tumors, thus supporting the antitumor immunity by activating CD8^+^ lymphocytes expressing IFN-γ, TNF-α, and granzyme B [66]. Also, treating WT mice bearing 4T1 tumors with anti-IL-1β followed by anti-PD-1 resulted in complete abrogation of tumor growth. Consequently, this highlights the importance of IL-1β as a master cytokine in tumor progression and thus could be a potential novel checkpoint for therapeutic targeting [66].

Another potential therapeutic target could be the metastasis- and inflammation-associated microenvironmental factor S100A4. In the aggressive triple-negative/basal-like subgroup, S100A4 was demonstrated to activate BC cells in various ways such as an increase in the secretion of pro-inflammatory cytokines, which, consequently, convert monocytes to macrophages. Another suggested mechanism would be promoting pro-tumorigenic activities such as stimulated epithelial–mesenchymal transition, proliferation, chemoresistance, and motility of cancer cells [63].

The enzyme 5-lipoxygenase (5-LO) is key for leukotrienes synthesis, which are potent pro-inflammatory lipid mediators involved in chronic inflammatory diseases including cancer [67]. In the presence of human MCF-7 BC cells, tumor-associated macrophages upregulated the proto-oncogene c-Myb inducing a stable transcriptional repression of the expression and activity of 5-LO eventually leading to attenuated T-cell recruitment and tumor progression [67]. Interestingly, the downregulation of 5-LO in TAMs required direct contact between macrophages and apoptotic cancer cells via Mer tyrosine kinase [67].

#### Role of Monocytes/Macrophages in Response to Therapy of Breast Cancer

Not only are monocytes and macrophages engaged in tumor progression or regression, but they also have a vital role in altering responses to treatment. In this regard, macrophage polarization and plasticity exhibited a role in the response of inflammatory breast cancer (IBC) cells to radiation therapy [68]. This was further supported by co-culturing polarized macrophages (M1 and M2) with IBC cells (SUM149, KPL4, MDA-IBC3, or SUM190) for 24 h followed by irradiation, after which the levels of IL4/IL13-induced activation of STAT6 signaling (phospho-STAT6 and Tyr641), and the expression of M2 polarization gene markers (CD206, fibronectin, and CCL22) increased [68]. Additionally, it was shown that, upon co-culturing with IBC cell lines, M1-polarized macrophages mediated radio-sensitivity, whereas M2-polarized macrophages mediated radio-resistance. This pattern was reversed in the presence of phosphopeptide mimetic PM37, which targeted the SH2 domain of STAT6 and prevented IL4/IL13-mediated STAT6 phosphorylating Tyr641 and consequently, decreased the expression of M2 polarization markers [68]. Moreover, proteomics analysis of IBC KPL4 cells using a kinase antibody array revealed induction of protein kinase C zeta (PRKCZ) in these cells following co-culture with M2-polarized macrophages. However, pretreatment by PM37 prevented and reversed radio-resistance when KPL4 cells were knocked-down using stable short hairpin RNA of PRKCZ, suggesting that PRKCZ is a potential target to inhibit M2 polarization and prevent radio-resistance [68].

Another study illustrated the role of macrophages in chemoresistance and detected the expression of a long noncoding RNA (lncRNA) that is associated with breast cancer brain metastases (BCBM) [69]. Lnc-BM was shown to increase JAK2 kinase activity and promote the expression of ICAM1 and CCL2 causing recruitment of macrophages into the brain [69]. In turn, the recruited macrophages produced oncostatin M and IL-6, thus enhancing BCBM [69]. Hence, such lncRNAs could be used as prognostic markers in BCBM patients and may be used as potential targets for therapy.

### 2.3. Role of Natural Killer (NK) Cells in Breast Cancer

NK cells are innate cytotoxic cells that possess a crucial role in defense against virally infected or transformed cells [70,71]. These cells are characterized by their phenotypic expression by being CD56^+^ CD16^+^ CD3^−^ cells that can be classified based on the expression of CD56 and CD16 molecules [42,72]. The two main subpopulations are CD56^bright^ CD16^-^ and CD56^dim^ CD16^+^. The former cells are cytokine producers, which are mainly found in lymph nodes and are less cytotoxic. In contrast, 90% of the circulating NK cells in the peripheral blood are CD56^dim^ CD16^+^ and are mainly involved in cytolytic activity of stressed cells via the release of perforin and granzymes [42,72].

Besides the direct killing of tumor cells, NK cells can indirectly boost the killing of the tumor cells via releasing cytokines and chemokines that cause the activation and recruitment of other immune cells [41,73]. Regarding solid tumors, the role of NK cells was first described in mice where NK cell depletion resulted in faster tumor growth after injection [74]. Other studies have shown that lower peripheral blood NK cells are associated with different solid tumors in patients with common variable immunodeficiency, suggesting a protective function of NK cells in certain tumor types [75,76]. In addition, the tumor microenvironment in lung cancer showed a decrease in the infiltration and activation of T and NK cells, indicating the importance of both cell types as target [77]. Both CD8 cytotoxic T and NK cells express a crucial activating receptor NKG2D that can trigger the release of their cytotoxic molecules to kill tumor and infected cells. In humans, NKG2D can bind to two types of ligands, firstly the major histocompatibility complex class I chain-related protein A (MICA) and MHC class I chain-related protein B (MICB) [78]. The other type is the unique long 16 (UL16)-binding proteins (ULBPs) 1–6, also known as retinoic acid early transcripts (RAETs) [78,79,80]. Upon transformation, tumor cells commonly downregulate MHC I molecules to evade cytotoxic T cell attacks. However, this would allow NK cells to detect and destroy tumor cells when the inhibitory signals are removed [81].

Typically, these ligands are absent in healthy tissues, with some expression in healthy gastrointestinal tract, bladder, and ureters [82]. The expression of MICA and MICB can be upregulated by stress stimuli such as heat shock, retinoic acid, or viral infections [83]. Additionally, the use of tumor hyperthermia as a tool in immunotherapy induced an upregulation of MICA, leading to an increase in the sensitivity of tumor cells to NK cell cytotoxicity [84]. CD56^+^ NK cells were investigated in different types of breast tissues, where they were found in breast lobules with fibrocystic changes. Moreover, the expression of MICA on stressed cells and the expression of CD56 on NK cells were negatively correlated with the severity of the abnormality [85,86]. In addition, there was a decrease in MICA expression in older people who are at high risk of developing breast cancer [87].

Breast cancer cells may express all the types of NKG2D ligands with variations in different individuals and stages of the disease [47]. Specifically, breast cancer patients with high expression of MICA and ULBP2 show better outcome with relapse free periods, that is further enhanced when both ligands are highly expressed in this type of tumor [88,89]. Currently, a wide array of therapeutic approaches is directed towards tackling the NKG2D/NKG2D ligand axis for better control and cure of cancer [90,91]. There are three different approaches of using this axis for immunotherapy: (1) to enhance the sensitivity of the cytotoxic lymphocyte receptor NKG2D; (2) to enhance expression of NKG2D ligands on tumor cells; and (3) to eliminate the soluble NKG2D ligands from tumor environment.

The activity and cytotoxicity function of NK cells is controlled by different cytokines such as IL-2, IL-12, IL-18, and IL-1, leading to upregulation of the expression of certain activating receptors [92,93,94]. Specifically, IL-2 cytokine has been utilized as an effective treatment for some malignancies and is currently used for immunotherapy. However, the use of IL-2 therapy has drawbacks such as toxicity, vascular leakage, and hypotension, along with other critical side effects [95,96].

In addition, there are many translational research data supporting the role of NK cells in breast cancer. In preclinical models, NK cells were shown to be critical in the anti-tumor responses mediated by HER2-targeting antibodies [97]. Further, trastuzumab induces ADCC, which leads to antigen release, cross-presentation by DCs, and increased NK cell activation and migration. In BC patients, increased NK infiltration into the TME has been observed upon treatment with HER2-targeting agents, supporting the notion that NK cells are important contributors to anti-tumor activity [97].

### 2.4. Role of neutrophils in breast cancer

Neutrophils are the most common innate immune cells in the body making up to 60–70% of the white blood cells. When activated, neutrophils can directly kill tumor cells via releasing reactive oxygen species (ROS) that can induce cell death by oxidative burst [98,99]. However, these cells are considered to be tumor promoting cells, in contrast to lymphocytes that are tumor suppressive cells. For this reason, the neutrophil to lymphocyte ratio (NLR) can be used as a predictive factor for tumor progression and overall survival in many types of cancer [100,101].

Moreover, neutrophils produce large amounts of proteases, cytokines, and chemokines that have cytotoxic effects on cancer cells [102,103]. However, the role of neutrophils in breast cancer is still controversial and not clear. Despite their low frequency in the primary tumor microenvironment, neutrophils play a major role in the initiation of the metastasis of breast cancer, where their numbers increase as the metastasis progresses [104,105]. They produce leukotrienes that have a tumorigenic effect on breast cancer cells migrating to the lungs, and thus promote seeding of these migrating cells in the lungs [36,106]. Additionally, some studies have shown that neutrophils inhibit CD8^+^ T cells and hence promote breast cancer cells seeding and metastasis [107]. However, other studies indicated that neutrophils produce hydrogen peroxide and nitric oxide that can inhibit breast cancer metastasis [108,109]. In addition, in murine breast cancer, neutrophils expressing CXCR2 have a major role in the lung metastasis of mesenchymal stromal cells (MSCs) [110]. This indicates that innate immune cells such as neutrophils can regulate the formation of a metastatic environment of breast cancer [111]. Another study showed that breast cancer cells metastasize by inducing systemic inflammation, via IL-1β induction of IL-17 secretion from (γδ) T cells in a mouse model. This IL-17 expression causes the release of G-CSF leading to systemic inflammation by activation and mobilization of neutrophils. However, tumor cells must inhibit CD8+ CTLs to have suppressive effects on neutrophils polarization. In addition, neutrophils activation and expansion increase pulmonary and lymph node metastasis without affecting primary tumor progression [107,112].

## 3. Use of Bioinformatics to Identify Genes Involved in Breast Cancer

With the advance technique of using bioinformatics and analyzing data from gene banks, it has been shown that certain genes are specifically involved in NK cells of breast cancer individuals. These genes are involved in the cellular mechanisms of the activating NK cells by breast cancer. For example, the protein encoded by ZAP70 gene is a very vital protein that is accompanied by immunoreceptor tyrosine activating motif (ITAM) that can be activated in NK cells [45]. This NK cell activation causes the release of granzyme A and B that are encoded by GZMA and GZMB genes. Additionally, the receptor of transforming growth factor TGF-β1, and its signal transducer (SMAD), are involved in the control of breast cancer cell growth and proliferation. In addition, FasL that belongs to the TNF superfamily, is involved in cell death via perforin-independent apoptosis pathways of target cells upon activation with TNF receptor members [113,114]. Indeed, downregulation of Fas in breast cancer patients correlate with short life expectancy [115]. Another gene that is included in the bioinformatics and is crucial in the activation of NK cells is HLA-E. HLA-E and HLA-G overexpression has been shown in HER2^+^ type of breast cancer [116]. HLA-E is a nonclassical MHC I that is expressed on tumor cells and binds to NK cells receptors such as CD94/NKG2A, CD94/NKGK2B and CD94/NKG2C [113,117]. In a study of 677 patients with early breast cancer, it has been shown that patients with lost classical HLA-I and expressing HLA-E and HLA-G show a shorter relapse free period, compared to those expressing classical HLA-I [116]. This suggests that there could be an interaction between these two classes of HLA molecules and that the presence of HLA-E/HLA-G with or without HLA-I can predict the outcome of relapse free period of breast cancer patients. HLA-E and HLA-F were also investigated in breast cancer and other solid tumors where it was observed that HLA-F expression positively correlated with tumor size and poor five-year survival rate [118,119].

## 4. Conclusions

Innate immune cells are important players in immune surveillance, tumor escape, and progression, besides their significant role in modulating the adaptive immune system. With the power of some new techniques such as bioinformatics, we improve our understanding regarding the crosstalk among innate immune cells and breast tumor microenvironment. Figure 2 shows the crosstalk among several cell types in BC microenvironment and demonstrates the effects of various soluble mediators released by these cells which either promote or suppress breast cancer cells. This may aid in choosing better immunotherapy approaches for better outcomes of breast cancer patients based on their immune system. However, the exact mechanisms of these innate cells remain unclear as new research explores their roles in breast cancer. Further, these therapeutic modalities are currently limited and are not always available in clinical practice.

Future studies may reveal better immunotherapeutic protocols for breast cancer with minimal side effects and harm to the patients. The high plasticity in the innate immune cells makes them more susceptible to tumor promoting molecules and cause their phenotypes toward type 2 anti-inflammatory response, except macrophages, which might be also be polarized towards type 1 pro-inflammatory cells in tumor microenvironment [120]. For these reasons, novel therapeutic modalities have been designed to reduce pro-tumoral activity while increasing anti-tumoral activity of innate immune system [121]. Based on the roles of innate immune cells in BC, we conclude that it is becoming crucial to characterize the immune system compartments in cancer patients. These members can have different presentation and accordingly, a unique signature that can be used in predicting survival and most importantly personalizing treatment to attain the best outcomes.

## Figures and Tables

**Figure 1 cancers-12-02226-f001:**
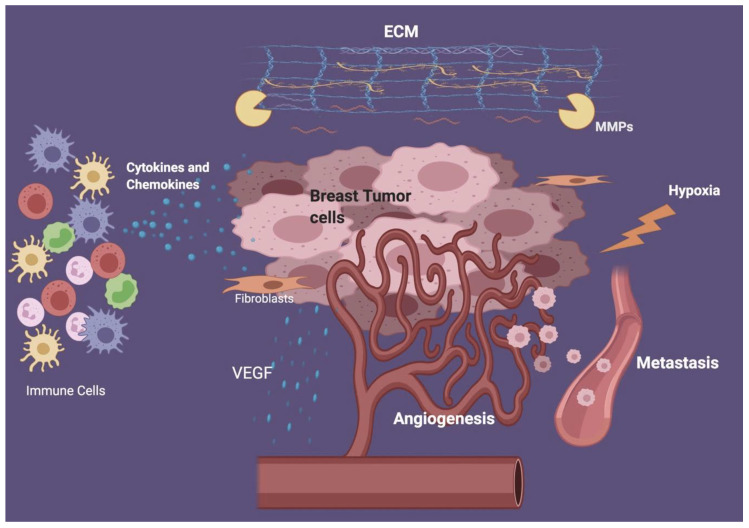
Breast tumor microenvironment (TME). Interaction among breast cancer cells and different processes involved in tumor microenvironment development. These include angiogenesis, tumor metastasis, ECM degradation and remodeling, cytokines and chemokines release, hypoxia, and VEGF release.

**Figure 2 cancers-12-02226-f002:**
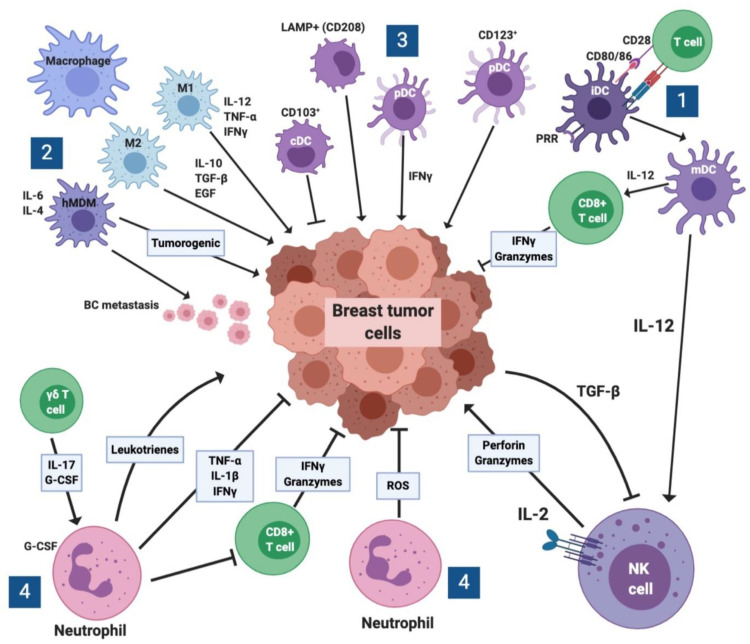
Interplay among various cell types and inflammatory molecules in breast cancer microenvironment. The cells can be divided into subcategories. (1) Role of killer cells. Upon the cognate interaction among T cells and immature dendritic cells (iDCs), the latter which also express pattern recognition receptors (PRRs) differentiate into mature dendritic cells (mDCs). These cells secrete IL-12 which activates cytotoxic T cells to release granzyme B and IFN-γ, which kill breast cancer cells. IL-12 also activates NK cells which release perforin and granzyme B to lyse tumor cells. Reciprocally, breast cancer cells utilize TGF-β to inhibit the function of killer cells. (2) Role of macrophages. Human monocyte-derived macrophages (hMDM) facilitate tumorigenesis by releasing cytokines. Similarly, M2 macrophages potentiate breast cancer cells growth through the release of suppressive molecules such as IL-10, TGF-β, and VEGF, which suppress the cytolytic activity of anti-tumor effector cells. In contrast, macrophages type 1 (M1) suppress tumor growth through the release of inflammatory IL-12, TNF-α and IFN-γ. (3) Role of dendritic cells. CD208 or LAMP^+^ DCs as well as plasmacytoid dendritic cells (pDCs) enhance tumor growth, whereas conventional dendritic cells (cDCs) inhibit their growth. (4) Role of neutrophils. Neutrophils may exert opposing effects on breast cancer cells growth. On the one hand, it may secrete reactive oxygen species (ROS) that may affect the growth of breast cancer cells. However, neutrophils by receiving IL-17 and G-CSF from γδ T cells may secrete leukotrienes, which inhibit tumor growth, or secrete the inflammatory cytokines TNF-α, IL-1β, or IFN-γ that could suppress tumor growth. Finally, neutrophils also inhibit the cytolytic activity of CD8^+^ T cells.

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
