# Peer review of "Understanding the Role of Innate Immune Cells and Identifying Genes in Breast Cancer Microenvironment"

_cancers, 2020, doi:10.3390/cancers12082226_

Round 1

Reviewer 1 Report

I read with very interest this review about the effects of innate immune cells on breast cancer. In general the manuscript is well written, easy to read (also for medicine and biologist students), figures are very nice and practice to understand this process. However we must consider that this mechanism is still little known and probably its knowledge may change over time. For example, based on what has been described, the possible therapeutic options are still too limited and in fact these targeted therapies are not available in clinical practice. It should be considered that the proposed therapeutic options are experimental and without substantial scientific relevance.

Author Response

We thank the reviewer for the comments We have added a sentence to the revised manuscript in the conclusion section, lines 403-405, according to the reviewer's suggestion

Reviewer 2 Report

This is a very good review of our current understanding of the innate immune cells, and of identifying genes, in the microenvironment of breast cancer. The introduction could be structured slightly differently, as upon first reading it appear a bit long (1.1-1.9). Might suggest (only a suggestion) that sections 1.4-1.9 of the introduction be titled as a separate section from 'introduction', and be a 2.1-2.6 section appropriately title so as to note the following sections to be those that address the individual groups of innate cells (no need to change the subsection titles as they are now).  That being said, though, as written, the divisions within it are well presented and certainly address the heterogeneity of breast cancer cells and the roles of the immune cells. This review does well in its emphasis of taking a closer look at the tumor microenvironment, and the effects of innate immune cells (to include monocytes, macrophages, NK cells, and neutrophils), and others. Each cell type is addressed in their individual sections within the introduction. Of special note is the section 1.4 (line 134) that addresses the all-important dendritic cells, and their subsets, and of their importance in breast cancer, which is very well presented.

Author Response

We thank the reviewer for the comments. Section 2: "Role of Innate Cells in Breast Cancer", has been added to the revised manuscript and the numberings of the subheadings have been changed accordingly.